# APTC-EC-2A: A Lytic Phage Targeting Multidrug Resistant *E. coli* Planktonic Cells and Biofilms

**DOI:** 10.3390/microorganisms10010102

**Published:** 2022-01-04

**Authors:** Karen Hon, Sha Liu, Sophie Camens, George Spyro Bouras, Alkis James Psaltis, Peter-John Wormald, Sarah Vreugde

**Affiliations:** 1Department of Health and Medical Sciences-Surgery, The University of Adelaide, Adelaide 5000, Australia; karen.hon@adelaide.edu.au (K.H.); sha.liu@adelaide.edu.au (S.L.); sophiecamens@outlook.com (S.C.); george.bouras@adelaide.edu.au (G.S.B.); alkis.psaltis@adelaide.edu.au (A.J.P.); peterj.wormald@adelaide.edu.au (P.-J.W.); 2Department of Surgery—Otolaryngology Head and Neck Surgery, Basil Hetzel Institute for Translational Health Research, The Queen Elizabeth Hospital, Woodville South 5011, Australia

**Keywords:** *E. coli*, antimicrobial, phage, stability, biofilm

## Abstract

*Escherichia coli* (*E. coli*) are common bacteria that colonize the human and animal gastrointestinal tract, where they help maintain a balanced microbiome. However, some *E. coli* strains are pathogenic and can cause serious infectious diseases and life-threatening complications. Due to the overuse of antibiotics and limited development of novel antibiotics, the emergence of antibiotic-resistant strains has threatened modern medicine, whereby common infections can become lethal. Phage therapy has once again attracted interest in recent years as an alternative treatment option to antibiotics for severe infections with antibiotic-resistant strains. The aim of this study was to isolate and characterize phage against multi-drug resistant *E. coli* isolated from clinical samples and hospital wastewater. For phage isolation, wastewater samples were collected from The Queen Elizabeth Hospital (Adelaide, SA, Australia) followed by phage enrichment as required. Microbiological assays, electron microscopy and genomic sequencing were carried out to characterize the phage. From the 10 isolated *E. coli* phages, *E. coli* phage APTC-EC-2A was the most promising and could lyse 6/7 *E. coli* clinical isolates. APTC-EC-2A was stable at a broad pH range (3–11) and could lyse the host *E. coli* at temperatures ranging between 30–50 °C. Furthermore, APTC-EC-2A could kill *E. coli* in planktonic and biofilm form. Electron microscopy and genomic sequencing indicated the phage to be from the *Myoviridae* family and of lytic nature. In conclusion, the newly isolated phage APTC-EC-2A has the desired properties that support its potential for development as a therapeutic agent against therapy refractory *E. coli* infections.

## 1. Introduction

The emergence of antibiotic-resistant human pathogens poses a severe threat to the health of the general population. One such pathogen is the gram-negative bacteria, *Escherichia coli* (*E. coli*), that belongs to the *Enterobacteriaceae* family, and resides harmlessly in the human gut along with other bacteria. However, some *E. coli* strains are pathogenic and responsible for various infectious diseases in humans and animals. These pathogenic *E. coli* can cause hospital- and community-acquired infections such as urinary tract infections (UTIs), gastrointestinal infections with diarrhea, meningitis, bacteriemia, pneumonia, surgical site infections, and sepsis [1].

Pathogenic *E. coli* is transmitted from infected human or animal feces to new susceptible hosts either via direct contact or via environmental reservoirs such as water, and soil [2]. A previous study by Maal et al. also revealed that the usage of treated wastewater as a resource for agricultural and industrial applications can be at the origin of infections [3]. Antibiotics are the current mainstay of treatment for *E. coli* infections; however, this can result in various well-documented side effects including a dysbiotic effect on an individual’s microbiome [4]. Furthermore, the overuse and misuse of antibiotics has also led to the emergence of multi-drug resistant strains to first-line antibiotics such as cephalosporins, fluoroquinolones and trimethoprim-sulfamethoxazole [5,6].

In the past few years, several reports showed the increment in resistance of two last-resort drugs (carbapenem and colistin) among *Enterobacteriaceae*, particularly in developing countries [7,8]. Discovering and developing new antibiotics is essential to combat this antibiotic resistance crisis, and yet it is genuinely challenging. The limited development of novel antibiotic drugs has prompted the search for alternative treatment options in recent years. Bacteriophage or phage therapy is one possible alternative for combating antibiotic-resistant bacterial infections. The attractiveness of their antibacterial activity in combating the presence of unwanted bacteria, including antibiotic resistant ones, without adverse effects on humans and animals has gained a lot of attention lately [9].

Phages are viruses specific to bacteria and are present in the same ecological niche of their host bacteria [10]. They are known as the most abundant entities in all known ecosystems such as aquatic and natural environments [11,12] and are found at high concentrations within humans and animals, especially in the gut. Phages that are released in the environment by humans and animals through defecation often persist longer than bacteria [13]. In this study, wastewater from hospitals was used for the isolation of *E. coli* phage because it is known to be an optimal source of the most effective therapeutic phages [14]. Several studies have indeed shown the success of phage isolation from wastewater that can target *E. coli* and used for phage therapy applications [15,16,17].

In this study, we describe the isolation and characterization of an *E. coli* bacteriophage APTC-EC-2A.

## 2. Materials and Methods

### 2.1. Bacterial Strains

The bacterial strains used in this study included *E. coli* XL 10 Gold (Agilent, Santa Clara, CA, USA) and clinical isolates Emergency Department (ED) 4, Palliative Care (PC) 22, Respiratory, Dermatology and Infectious Ward [18] 14, Orthopaedic and Plastic Surgery Ward (OPS) 13, isolated from non-treated effluents from respective departments from The Queen Elizabeth Hospital (TQEH) (Woodville, SA, Australia). The *E. coli* 378 and 452 were clinical isolates kindly donated by Dr. Rietie Venter (School of Pharmacy and Medical Sciences; University of South Australia).

### 2.2. Wastewater Collection and Process

A total of 200 mL of wastewater samples was harvested from the following five departments: Emergency Department (ED), Palliative Care (PC), Respiratory, Dermatology and Infectious Ward [18], Orthopaedic and Plastic Surgery Ward (OPS) and Intensive Care Unit (ICU) (all from TQEH) at 5 different time points. The wastewater was stored at 4 °C and processed within 2 days. Briefly, 50 mL of wastewater samples were centrifuged at 10,000× *g* for 30 min at 4 °C and supernatants were filtered through a 0.22 μm syringe filter (PALL Acrodisc, Port Washington, NY, USA) and the clarified supernatants were stored at 4 °C for further use.

### 2.3. E. coli Strain Isolation from Wastewater

A total of 100 µL of well-mixed pre-processed wastewater samples were diluted into sterile MilliQ water with serial 10-fold dilutions. Then, each dilution of 100 µL was inoculated onto MacConkey-Sorbitol agar and Blood agar plates (Oxoid, Thebarton, Australia) using an inoculation spreader. Plates were incubated at 37 °C overnight. Individual colonies were selected and identified using MALDI-TOF system (Bruker, Preston, VIC, Australia). The bacteria were grown on either 1.5% Luria-Bertani agar (LB) or in LB (Oxoid, Thebarton, Australia) broth at 37 °C for approximately 16 h depending on the experiment.

### 2.4. Antibiotic Sensitivity Testing

The minimum inhibitory concentration (MIC) of 11 different antimicrobial agents against the *E. coli* strains was assessed using the broth microdilution method as described [19] with modifications. Antibiotics tested were amoxicillin-clavulanate, ampicillin, ciprofloxacin, cefoxitin, doxycycline, gentamicin, imipenem, ertapenem (all from Chem-Supply, Gillman, Australia); streptomycin, penicillin, and trimethoprim (all from Sigma-Aldrich, Castle Hill, NSW, Australia). In brief, single colonies of *E. coli* were suspended in 0.9% saline and adjusted to 0.5 MFU and diluted 1:100 in LB broth in serially diluted antibiotics (range from 128 to 0.25 µg/mL) in 96 well microtitre plates. The last two rows of the plates were set up as positive control (without antibiotic and inoculated with bacterium) and a negative control (without antibiotic and bacterium). The plates were then incubated for 24 h at 37 °C. The MICs for the various *E. coli* strains tested were determined based on the Clinical and Laboratory Standards Institute (CLSI) guidelines, National Antimicrobial Resistance Monitoring System (NARMS) guideline or the European Committee on Antimicrobial Susceptibility Testing (EUCAST). Isolates with MICs above the susceptible breakpoint were classified as non-susceptible.

### 2.5. E. coli Phage Isolation Using a Double Layer Agar Assay

Phage was isolated using a double layer agar assay with modifications [20]. Phage isolation was achieved by incubating 500 µL of filtered supernatant with 100 μL of an overnight *E. coli* XL10 Gold bacterial culture for 10 min, followed by mixing with 4 mL of molten top agar (0.4% *w*/*v* LB agar). Then, the mixture was immediately overlaid into 1.5% LB agar plates to create a bacterial lawn and allowed to solidify for 10 min. The plate was incubated overnight at 37 °C for plaque formation. Individual plaques with different morphologies were picked from the overlay using a sterile pipette tip and eluted into 2 mL glass vials (ThermoFisher Scientific, Sydney, NSW, Australia) containing SM buffer (NaCl, MgSO_4_.7H_2_O and 1 M Tris-Cl, pH 7.5) at 4 °C for a few days to allow phage particles to diffuse from the soft agar.

### 2.6. Phage Isolation Using Enrichment Culture

If no plaque was observed, the enrichment method was used as an alternative way to isolate potential phages. A total of 100 μL of overnight *E. coli* XL10 Gold or isolated *E. coli* from the wastewater sample bacterial broth culture and 1 mL of filtered crude lysate were added into 10 mL of LB broth. The mixture was incubated overnight at 37 °C with agitation (180 rpm) for 24 h. Then, the mixture was centrifuged at 10,000× *g* for 10 min and the supernatant was filtered through a 0.22 μm syringe filter to obtain phage lysate. The double layer agar assay was performed as previously described to isolate phage after enrichment.

### 2.7. Purification of Phage

Isolated phage was purified using the double layer agar assay. Then, 10 µL of phage suspension in sodium magnesium buffer (SM buffer: 100 mM sodium chloride (Oxoid, Thebarton, SA, Australia); 8 mM magnesium sulfate heptahydrate (Sigma-Aldrich, Castle Hill, NSW, Australia); 50 mM 1M Tris HCL, pH 7.5) were mixed with 100 μL of an overnight culture of bacterial host for 10 min. The mixture was added to 4 mL of 0.4% LB agar and then poured over 1.5% LB agar plates. The plates were incubated overnight at 37 °C. The procedure was repeated three times to obtain single plaques with clear and homogeneous plaque morphology.

### 2.8. Phage Propagation and Phage Titration

All phages were propagated by inoculation with exponentially growing *E. coli* XL 10 Gold bacterial culture. LB broth was inoculated with 1% overnight bacterial culture and incubated at 37 °C in a shaker with agitation (180 rpm) for an hour. Then, phage was added into the culture at multiplicity of infection [18] of 1. After overnight incubation at 37 °C with 180 rpm agitation, 2% chloroform was used to lyse the bacteria. The cellular debris was then cleared by centrifugation (30 min, 10,000× *g*, 4 °C) and the supernatant was filtered through 0.22 μm syringe filter.

Phage titres were enumerated using the double layer agar method. Aliquots of 100 μL from each 10-fold serially diluted lysate was mixed with 100 μL of bacterial overnight culture and 4 mL of 0.4% top agar was added to the tubes. The mixtures were poured onto LB agar plates and allowed to solidify for 10 min. Plates were incubated at 37 °C for 24 h. Phage titration was performed in triplicate and calculated. The phage lysates were then aliquoted and stored at 4 °C for further testing.

### 2.9. Host Range

The phage host range was determined using spot tests as described [21]. Bacterial *E. coli* lawns were prepared using a double-layer LB agar and 3 μL purified phage suspension (~10^10^ pfu/mL) were spotted onto the upper layer medium after serial dilution and left to incubate overnight. Plaque clarity on the bacterial lawn was differentiated into three categories: a clear plaque (sensitive (+)), turbid plaque (semi-sensitive (+/−)) or no plaque (resistant (−)) respectively [21]. Thus, phages that displayed a zone of lysis against most isolates were selected for further studies.

### 2.10. Transmission Electron Microscopy (TEM)

For this process, 10 µL of sample (~10^10^ pfu/mL) was added to 90 µL of SM buffer, and 5 µL of diluted sample was placed on the carbon/formvar coated grid (ProSciTech Pty Ltd., Kirwan, QLD, Australia) for 3 min. Grid was wicked dry with filter paper and fixed with 5 µL of EM fixative (1.25% glutaraldehyde, 4% paraformaldehyde in PBS to which 4% sucrose had been added) for 2 min. Then, the grid was washed with distilled water, followed by the addition of 5 µL of 2% uranyl acetate as the negative stain. Excess liquid was wicked off with filter paper after 2 min. The sample was then visualized under FEI Tecnai G2 Spirit 120 kV TEM (FEI Technologies Inc., Hillsboro, OR, USA).

### 2.11. Phage Stability Assay: pH, Temperature and Long-Term Storage

The pH stability was examined by pre-incubating the phage lysate with an SM buffer of different pH levels (3, 4, 5, 6, 7, 8, 9, 10, 11 and 12 respectively). Briefly, phage (~10^8^ pfu/mL) was suspended in SM buffer with pH ranging from 3 to 12 (*v*/*v* = 1:1) and incubated at room temperature for 1 h. For long term storage stability, 1 mL stock solution of each phage was aliquoted and stored at room temperature, 4 °C and 80 °C for 2 weeks, 1-, 2-, 3-, and 4–months in LB broth. For thermal stability, phage lysates of known titre were incubated at 30 °C to determine the effects of different temperatures on the stability of phage after an hour of incubation. The procedure was repeated at temperatures 4 °C, 37 °C, 40 °C, 50 °C, 60 °C, 70 °C and 80 °C. Phage titers were determined to evaluate their stability at different conditions using double layer agar method described above. Each of the studies was repeated three times.

### 2.12. Inhibition Assay (Bacteriolytic Characteristic of the Phage)

Overnight *E. coli* cultures were inoculated at 1:100 in fresh medium to achieve an OD value of 0.2. Phage was added at MOIs of 0, 0.1 and 1 to the diluted bacterial culture and incubated at 37 °C in a shaker with agitation (180 rpm). The absorbance OD600 nm (Biorad Smartspec^TM^ 3000, Memphis, TN, USA) of the culture broth was measured every 30 min after the onset of incubation for a period of 4 h. Inhibition assays were performed three times.

### 2.13. Biofilm Quantification by Crystal Violet Staining

A 1 McFarland unit *E. coli XL10* suspension in saline was diluted 1:15 into LB broth and gently mixed by inversion. The wells adjacent to the edge of a clear polystyrene 96-well plate (Costar^®^, Corning Incorporated, Corning, NY, USA) were filled with 200 μL sterile PBS, and a column of six wells was filled with only LB broth, used as a negative control, and the rest of the wells were filled with 150 μL of the bacterial suspension. The plate was then incubated for 48 h on a gyratory mixer (Ratek, VIC, Australia) at 37 °C to allow biofilm development.

After 48 h, the liquid contents of the bacterial wells were gently aspirated, followed by washing twice with sterile PBS to remove all the planktonic bacteria. The phage in LB broth at concentration of 5 × 10^8^ pfu/mL and 5 × 10^9^ pfu/mL were applied. In each plate, there was LB broth as negative control and LB with bacterial suspension as positive control. Each treatment of 200 μL was plated, and the microplates were incubated at 37 °C.

After 24 h post-treatment, the liquid contents of the bacterial wells were gently aspirated, followed by washing twice with sterile PBS to remove all the planktonic bacteria. Then, the plate was stained with 200 μL/well 0.1% crystal violet (Sigma-Aldrich, Castle Hill, NSW, Australia) for 15 min. The stained plates were rinsed by two rounds of gentle immersion into distilled water and left to dry overnight. The crystal violet stain was eluted by the application of 200 μL/well 30% acetic acid and the plate was incubated on a shaker at room temperature for an hour. Absorbance at 595 nm was measured for each well using the Fluostar Optima microplate reader (BMG Labtech, Ortenberg, Germany), with 200 μL/well 30% acetic acid in stained well of LB broth only as blanks.

The OD595 was measured in the microplate reader and the CV-stained wells as described above. The relative biomass was determined by normalizing to the absorbance of the positive control. The whole procedure was repeated with ED, PC and 378 bacterial strains.

### 2.14. LIVE/DEAD Staining

Biofilm viability was determined using a LIVE/DEAD *Bac*Light Bacterial Viability Kit (Thermo Fisher Scientific, Eugene, OR, USA). Briefly, biofilms were grown for 48 h in 8-well cell imaging slides (Eppendorf, Hamburg, Germany) containing no bacteriophage as a positive control. Wells were aspirated and washed gently twice with 0.9% saline before application of 350 μL/well of APTC-EC-2A treatment in LB at concentrations of 1 × 10^8^ and 10^9^ pfu/mL. Following the 24 h incubation, wells were aspirated and washed again prior to application of 180 μL/well 5% glutaraldehyde fixative (Sigma-Aldrich, Castle Hill, NSW, Australia) for 45 min. Prior to staining, wells were aspirated and washed again. Two stock solutions of stain SYTO9 and propidium iodide [22] were each diluted in MilliQ water to give a 1 mL:1.5 µL:1.5 µL ratio and 400 µL applied to each well for 15 min before aspirating and washing wells. Live SYTO9-stained cells and dead PI-stained cells were visualised with a confocal laser microscope at 20X magnification, and the percentage of cell death was determined (Zeiss LSM700, Carl Zeiss AG, Oberkochen, Germany).

### 2.15. Genomic Extraction, Sequencing and Analysis

Genomic sequencing was performed for APTC-EC-2A. DNA extraction from a 1 mL aliquot of phage (>1 × 10^10^ pfu/mL) was performed using the Phage DNA Isolation Kit (Norgen Biotek Corp., Thorold, ON, Canada), following the protocol provided by the manufacturer. Genomic sequencing of the isolated DNA was conducted by an external laboratory (SA Pathology, Adelaide, Australia). Paired-end sequence reads were obtained on an Illumina sequencer, yielding paired end 150 bp reads. Sequencing reads were quality controlled using FastQC (v 0.11.9) [23], and trimmed with Trimmomatic [24].

### 2.16. Genomic Analysis

The genome was assembled using Unicycler v 0.4.8 [25] into one single circular contig and annotated with MultiPhATE2 [26]. Anti-microbial resistance (AMR) and virulence genes were identified by screening the assembled genome through the Comprehensive Antibiotic Resistance Database (CARD) (https://card.mcmaster.ca/; accessed on 8 September 2021) and virulence factor database (VFDB) (http://www.mgc.ac.cn/VFs/; https://card.mcmaster.ca/; accessed on 8 September 2021) by using ABRicate (v 1.0.1) (https://github.com/tseemann/abricate; https://card.mcmaster.ca/; accessed on 8 September 2021). The whole genome visualisation of APTC-EC-2A was created using BRIG v 0.95 [27].

The phages with the highest similarity to APTC-EC-2A in the NCBI nucleotide (‘nt’) database were determined using BLASTn v 2.9.0+ [28]. Genome alignments of APTC-EC-2A and the top BLAST hits were obtained using Mauve [29] and plotted in R using genoplotR [30]. To create the phylogenetic tree based on the terminase large subunit sequences, all of the top phage genomes (APTC-EC-2A and the top BLAST hits) were re-annotated using Prokka v 1.14.6 [31]. The query_pan_genome command from Roary v 3.13 [32] was then used to extract all the terminase large subunit sequences. A multiple sequence alignment was created using Clustal Omega v 1.2.8 [33]. Based on this alignment, a maximum likelihood phylogenetic tree was created using IQTree v 2.0.3 [34], specifying 1000 ultrafast bootstrap replicates.

### 2.17. Statistical Analysis

Statistics were analysed using the GraphPad Prism v.9 (GraphPad Prism, version 9.0.0 (86); Macintosh Version Software MacKiew© 2022–2020 GraphPad Software, LLC.; San Diego, CA, USA). Statistical significance for all results was analysed using one-way analysis of variance (ANOVA) and Dunnett’s multiple comparisons test. Significance was determined at a *p*-value < 0.05. All experiments were repeated three times and performed in triplicate or six replicates. The mean values of the replicates were obtained with standard deviation (SD).

## 3. Results

### 3.1. Bacterial Isolation from Wastewater

More than 20 species of bacteria were isolated with *E. coli* present in the wastewater of all five departments. *E. coli* was found to be most abundant in the wastewater from the RDI department (47% of total bacteria present), followed by PC (10%) and ICU (9.3%) departments. Representative *E. coli* strains isolated from each of those departments were used in this paper. Figure 1 depicts the various bacteria isolated from the five departments in The Queen Elizabeth Hospital (TQEH) wastewater.

### 3.2. Determination of Antimicrobial Susceptibility for E. coli Strains

All the *E. coli* strains were multi-drug resistant, demonstrating resistance to more than three antimicrobial classes frequently used for the treatment of *E. coli* infections. The antibiotic resistance profile of six *E. coli* strains (two clinical isolates and four from wastewater) are shown in Table 1.

### 3.3. Isolation of Phages with E. coli XL10 Gold

Wastewater samples were screened separately for the presence of *E. coli* phages using *E. coli* XL10 Gold strains. A total of ten phages were detected in wastewater from four different departments from TQEH using the double-layer method (Table 2 and Table 3). Plaque morphology for each of those phages along with their titres is shown in Table 2. For the emergency department, no plaques could be observed in both the double-layer and enrichment method. The bacteriophages were then selected for further purification and characterization.

### 3.4. Phages Host Range Analysis

The isolated phages were tested for their lytic activity against a total of 7 *E. coli* strains. These included 1 lab strain (*E. coli* XL 10 Gold), 4 strains isolated from wastewater and 2 clinical isolates. *E. coli* XL 10 Gold was susceptible to all 10 phages tested. LB agar plate with a representative plaque indicating sensitive, semi sensitive, and resistant to phage is shown in Appendix A. 4 of the *E. coli* strains were semi-sensitive to phages APTC-EC-8A and APTC-EC-8B. APTC-EC-2A was able to lyse 6/7 *E. coli* strains. Among these phages, APTC-EC-2A demonstrated the broadest spectrum of activity and therefore was selected for further characterization. Results are detailed in Table 4.

### 3.5. Characterisation and Stability of Selected Phage

#### 3.5.1. Morphology of Phage

The transmission electron microscopy showed that phage APTC-EC-2A had an icosahedral head (about 88 nm diameter) with a long contractile tail (of ~47 nm), a base plate and several tail fibers (Figure 2). The overall morphology indicated that this phage was a T4-like phage, belonging to the *Myoviridae* family.

#### 3.5.2. Phage Stability: Long-Term Storage, Temperature and pH

Phage was tested for storage stability and was found to be more stable at a refrigeration temperature (4 °C) than at room temperature and a frozen temperature (−80 °C) over a period of 4 months. At 4 °C and −80 °C, minimal reductions in phage titres were observed, whereas there was a gradual reduction in phage titre at room temperature over time, as shown in Figure 3a.

A thermal stability test was performed at various temperatures for phage APTC-EC-2A (Figure 3b). It was found that APTC-EC-2A was stable when exposed to temperatures between 4 and 50 °C for 1 h, as no significant reduction in phage titres were detected. However, at 60 °C, phages showed a reduction in titres of up to 52%. At 70 °C and 80 °C, phage titres were below detectable levels as no plaques were observed. Therefore, our results showed that phage APTC-EC-2A was sensitive to exposure to high temperatures (70 °C to 80 °C) and could withstand exposure to lower temperatures for 1 h (4 °C up to 50 °C).

The effect of various pH on phage stability was also evaluated (Figure 3b). The APTC-EC-2A phage showed great stability when incubated under a range of pH values from pH 3 to pH 11, with no significant reductions in phage titres observed. However, phage APTC-EC-2A showed 100% inactivation at pH 12.

### 3.6. Inhibition Assay

The bacteriolytic characteristics of APTC-EC-2A against various *E. coli* bacterial strains (XL 10 Gold, ED, PC, and 378) were tested at various MOI and compared to an untreated control over 3.5 h. The growth of the bacterial strains without phage infection (MOI = 0) followed a typical sigmoid growth curve. The growth of all four strains was suppressed in the presence of phage APTC-EC-2A at MOI = 0.1 and MOI = 1 after incubation for 3.5 h (*p* < 0.001). Results are shown in Figure 4.

### 3.7. Phage Reduces the E. coli Biofilm Biomass and Viability

To investigate whether APTC-EC-2A could also decrease the biomass and viability of 48-h established *E. coli* biofilms, crystal violet and Live/Dead viability assays were performed after treatment with phage APTC-EC-2A for 24 h. Phage APTC-EC-2A, at a concentration of 5 × 10^9^ pfu/mL, significantly reduced the biomass of XL10 Gold, ED, PC and a further 378 strains compared to the no-treatment control. A higher phage concentrations had a more profound effect on reducing the biofilm biomass than lower concentrations, reducing the biofilm biomass by up to 53% (range 16–54%) when treated with APTC-EC-2A at the concentration of 5 × 10^8^ pfu/mL; and up to 70% (range 27–70%) when treated with APTC-EC-2A at the concentration of 5 × 10^9^ pfu/mL (Figure 5a).

The LIVE/DEAD *Bac*Light Bacterial Viability Kit was used to determine the percentage of live (SYTO9, green) and dead (PI, red) *E. coli* cells after APTC-EC-2A phage treatments. There was a statistically significant increase in the percentage of dead cells of *E. coli* XL10, ED and PC in APTC-EC-2A-treated biofilms as compared to the untreated control at both phage concentrations. However, for *E. coli* 378, only the highest concentration of APTC-EC-2A showed a significant percentage of cell death compared to the control (Figure 5b).

### 3.8. Genomic Extraction, Sequencing and Analysis

The full nucleotide sequence of the *E. coli* phage APTC-EC-2A genome was determined (NCBI accession number: OK274152). The genome of APTC-EC-2A was 166,608 bp in length, with a GC content of 35.54% (Figure 6). The annotation information, such as position, direction and functions of each gene and conserved proteins are summarized in Appendix A.

The bioinformatic analyses of the APTC-EC-2A genome indicated that it contains 281 predicted Open Reading Frames (ORFs), of which 45 ORFS were located on the forward strand of the DNA along with 11 tRNAs, while 236 ORFs were located on the complementary strand (Figure 6 and Appendix A). The genetic screening of APTC-EC-2A against the Virulence Factor Pathogenic Bacteria (VFPB) database suggested APTC-EC-2A to be free of lysogenic genes or virulence-associated, toxic or antibiotic resistance genes. Furthermore, the APTC-EC-2A genome was found to be devoid of integrases.

According to similarity searches of the NCBI nucleotide (‘nt’) database using the BLASTn algorithm, the APTC-EC-2A phage has a nucleotide identity ranging between 96 and 99% to various phages in the Myoviridae family, specifically of the species Tequatrovirus. The top 10 phages with the highest identity to APTC-EC-2A are detailed in Table 5. Among these 10, there are 7 Escherichia phages, 1 Yersinia phage, 1 Shigella phage and 1 Enterobacter phage.

Figure 7a shows the synteny plot of APTC-EC-2A with the top 10 BLASTn hits. The segment between 75 kbp and 130 kbp appears to be shared amongst the various phages and this segment of the genome corresponds to structural proteins including the baseplate and tail fibers. The most notable divergent areas are located between 0 and 15–20 kbp which consist of hypothetical proteins and spackle proteins, and at 65–75 kbp which contains many hypothetical proteins of unknown function and tRNAs. A maximum-likelihood phylogenetic tree for APTC-EC-2A was also generated using the top 25 BLAST hits, based on the amino acid sequences of the phages’ terminase large subunits. The tree shows that the terminase of phage APTC-EC-2A is identical to the terminase of the Escherichia phage vB_EcoM_G4498, Escherichia phage vB_EcoM_G29 and Escherichia phage vB_EcoM_Shinka.

## 4. Discussion

In this study, a total of 10 phages were successfully isolated from wastewater effluents from various departments at TQEH, with *E. coli* XL10 as their host. Only one of such phages, APTC-EC-2A, was selected for full characterisation. We found that APTC-EC-2A had a broad host range, demonstrating lytic properties against five out of six multidrug-resistant *E. coli* isolated from wastewater and patients. APTC-EC-2A was active against both planktonic and biofilm forms and was stable at various temperatures up to 50 °C. Genomic sequencing and TEM indicated APTC-EC-2A to be of lytic nature, belonging to the *Myoviridae* family.

*E. coli* is a ubiquitous and typically harmless commensal, but there are also pathogenic strains belonging to this species. These can cause urinary tract infection, diarrhea, meningitis and sepsis in both children and adults [35]. A report published by the Australian Commission on Safety and Quality in Health Care shows an increasing number of pathogenic *E. coli* have emerged to be resistant to antibiotics and such reports have played a role in the revival of interest in phage treatments. *E. coli* phages are not limited to use in phage therapy applications, as they can also be used for microbiota treatment, food antimicrobial protection as well as environmental control.

For instance, phages have been used widely to protect the food chain and livestock against foodborne pathogens including Shiga toxin-producing *E. coli* [36,37]. Several studies have demonstrated the efficacy of *E. coli* phage against planktonic and biofilm forms of *E. coli* [15,38,39]. In another study, an *E. coli* phage cocktail demonstrating promise for reducing an *E. coli* infection efficiently without disrupting the gut microbiota in vivo [40]. Therefore, more *E. coli* phages are needed for their potential use in those various indications. To date, limited information is available on *E. coli* phage general characteristics and their physiology. Additionally, the stability of phages under various stress conditions has not been well documented.

Theoretically, tailed phages use receptor-binding proteins (RBP) at the distal end of the tail for phage-host interaction in which tail spikes, tail fibers or spike proteins function as RBPs by attaching the phage to the host-cell surface [41]. For Gram-negative bacteria, such as *E. coli*, a variety of surface-exposed components can be exploited as phage receptors, including outer membrane proteins (OMPs), lipopolysaccharides (LPS), pili and flagella [42]. Phages that target a highly specific bacterial receptor therefore often have a narrow host range whilst phages that target multiple receptors typically have a broader host range [43]. This could potentially explain our finding of a narrow host range for nine of the isolated phages lysing only one or a small number of strains. In contrast, APTC-EC-2A, which could lyse most of the strains tested, could potentially harbour various RBPs, thereby expanding the host range of that phage. Further experiments are required to test this hypothesis.

According to the morphological characteristics evaluated by TEM, APTC-EC-2A likely belongs to the *Caudovirales* order. In general, *Caudovirales* are divided into three families depending to the tail morphology, as *Myoviridae* have long contractile tails, *Siphoviridae* have long non-contractile tails and *Podoviridae* have short non-contractile tails [44]. Our genomic analysis was in line with the TEM report and indicated that APTC-EC-2A to belong to the *Myoviridae*. Genomic analysis also indicated the strictly lytic nature of APTC-EC-2A, which was furthermore supported by the lytic clearance plaques produced by APTC-EC-2A. Additionally, the absence of genes coding for toxins and other virulence factors that might affect eukaryotic cells makes phage APTC-EC-2A a good candidate for phage therapy or biotechnology applications.

The stability of phages at various temperatures and thus, the potential long-term storage of phages, is critically important [45]. Over a storage period of 4 months, APTC-EC-2A maintained its viability at room temperature, 4 °C and −80 °C. A minimal activity loss was demonstrated by APTC-EC-2A between 4 °C and −80 °C which showed that phage survives well at these conditions. A significant reduction in viability however was observed at room temperature which is in line with previous studies which suggested the storage of phages at RT is recommended for a short period of no longer than 40 days [46].

Physico-chemical factors also play an important role in affecting the stability and infectivity of phage and temperature is one of them. Several studies have shown that phage inactivation is mainly due to the nucleic acid and protein denaturation caused by high temperatures [46,47]. Another study showed that temperature affects the morphology of phages; whereby the exposure to high temperatures leads to the denaturation of viral capsid, which resulted in the release of phage DNA, decomposition of the head and tail structure of phage as well as an aggregation of the phage tail [48]. This study was in line with those observations as a drastic loss of phage activity was seen at temperatures of 60 °C and above. APTC-EC-2A was however highly stable at temperatures in the range of 30–50 °C, supporting the potential use of this phage in humans.

Additionally, APTC-EC-2A was stable at a broad pH range (3–11) compared to the phages in other studies [46,49,50,51]. Jończyk et al. revealed that phage titers of T4 bacteriophage decreased by half at pH 9.2 and no activity was detected at pH 4 after an hour of incubation. However, APTC-EC-2A was found to be stable at a low pH, which is particularly important for *E. coli* phages because of the acidic nature of the stomach, which would need to be transited by any oral *E. coli* phage formulation targeting intestinal *E. coli* infections. On the other hand, exposure to high alkalinity conditions such as pH12 can inactivate phage by acting on phage’s infection cycle and influencing the ability of phage to reproduce [46]. Previous work also suggested that the high concentration of hydrogen and hydroxyl ions in extreme pH conditions dominates phage inactivation mechanisms, resulting in the direct oxidation of the phage surface (capsid, tail fibers, etc), followed by dissociation of the capsid [47,52,53]. Overall, our results indicate that phage APTC-EC-2A can survive in relatively extreme conditions.

The efficiency of bacterial lysis is regulated by MOI and the results showed that phage APTC-EC-2A can efficiently reduce the growth of *XL10*, ED, PC and 378 bacterial strains at a higher MOI. This result is consistent with previous studies that revealed that the bacterial-growth reduction increases or occurs earlier when the MOI is higher [54].

According to the sequencing analysis, the genome of APTC-EC-2A showed approximately 96–99% similarity to the genomes of the top ten most homologous BLAST hits. These included *E. coli* phages, as can be expected, but also *Yersinia* and *Shigella* phages. However, the APTC-EC-2A phage is distinguished from closely related phages by the presence of divergence areas at 15–20 and at 65–75 kbp. Between the divergence areas, spackle proteins are present, which play a role in inhibiting gp5 lysozyme activity, thereby preventing the DNA injection of phage into the host cytoplasm, and as a consequence cause bacteria to become infected with phage APTC-EC-2A, becoming resistant to later infection by APTC-EC-2A phage or closely related phages [55].

Interestingly, the areas of highest similarity between those phages corresponded to genomic-sequences-encoding structural proteins, such as the baseplate and tail fibers. Since these genes encode RBPs, these findings might indicate that *Yersinia* and *Shigella* bacterial strains are sensitive to phage APTC-EC-2A. Further experiments are required to test this hypothesis.

## 5. Conclusions

In conclusion, all of the properties tested, including broad host range, strong lytic activity and stability at a wide range of temperatures and pH levels exhibited by APTC-EC-2A, make it an attractive phage for further preclinical and clinical development.

## Figures and Tables

**Figure 1 microorganisms-10-00102-f001:**
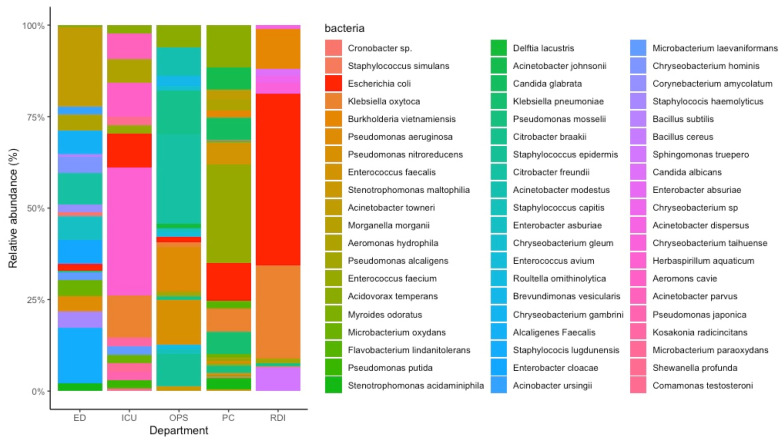
Isolated *E. coli* from wastewater from The Queen Elizabeth Hospital (TQEH). The percent stacked column bar chart indicates the relative abundance of isolated bacteria from each department in TQEH. *E. coli* is indicated in red. TQEH = The Queen Elizabeth Hospital; ED = Emergency Department, ICU = Intensive Care Unit, OPS = Orthopaedic and Plastic Surgery Ward, PC = Palliative Care and RDI = Respiratory, Dermatology and Infectious Ward.

**Figure 2 microorganisms-10-00102-f002:**
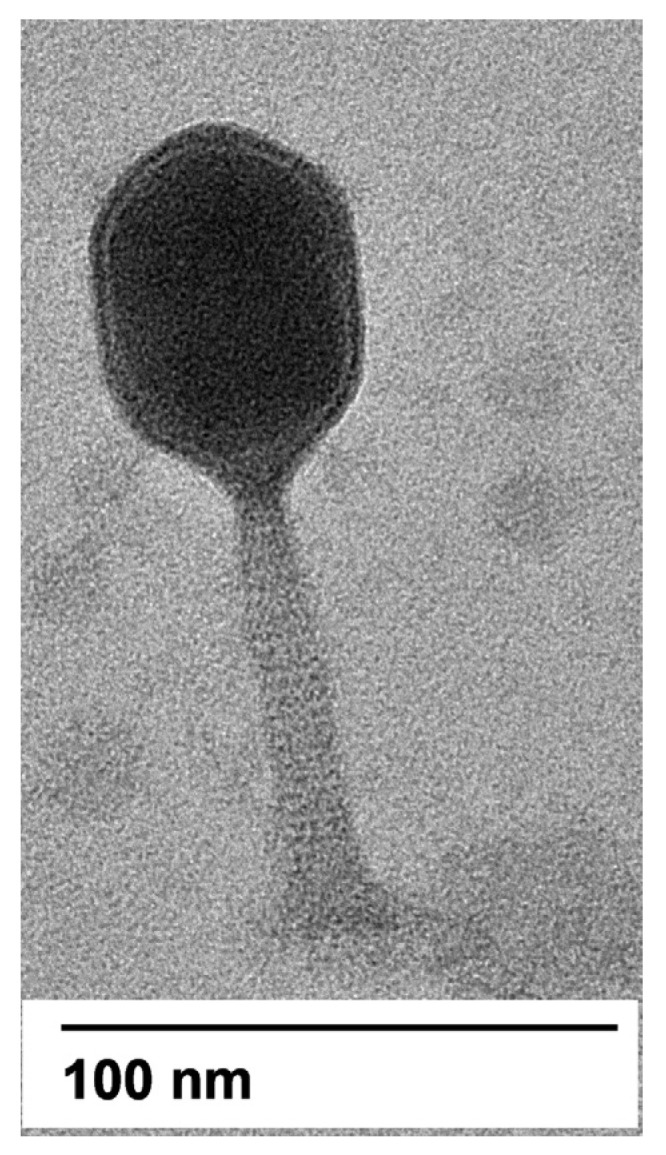
Electron micrograph of APTC-EC-2A. Transmission electron micrograph of *E. coli* APTC-EC-2A phage, belonging to the *Myoviridae* family. Scale bar represents 100 nm.

**Figure 3 microorganisms-10-00102-f003:**
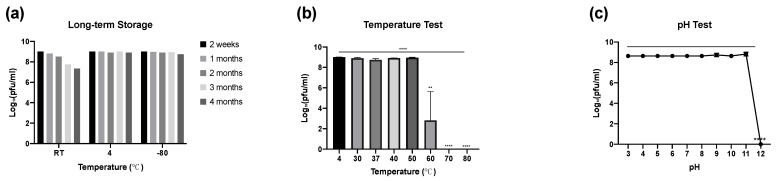
Stability Test for APTC-EC-2A. Graphs (log10 pfu/mL) showing the (**a**) long-term storage stability of phage APTC-EC-2A at room temperature, 4 °C and −80 °C for up to 4 months, (**b**) phage thermal stability at temperatures ranging from 4 °C to 80 °C, (**c**) phage pH stability test with pH ranging from pH = 3 to pH = 12. Significance was determined by one-way ANOVA with Dunnett’s multiple comparisons post-hoc test. Bars represent standard deviation (SD). **, *p* <0.001; ****, *p* < 0.0001. RT = room temperature.

**Figure 4 microorganisms-10-00102-f004:**
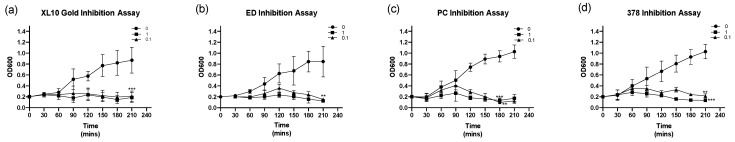
Kinetic of bacterial growth in the presence of phage APTC-EC-2A. Four bacterial strains: (**a**) XL10 Gold, (**b**) ED, (**c**) PC, and (**d**) 378 were tested against APTC-EC-2A at the concentration of 0 (without phage), 1 and 0.1 MOI. Error bars represent the standard deviation (*n* = 3) for 240 min. Significance was determined by one-way ANOVA with Dunnett’s multiple comparisons test in comparison with untreated biofilms; *p* < 0.01 (**) and *p* < 0.001 (***).

**Figure 5 microorganisms-10-00102-f005:**
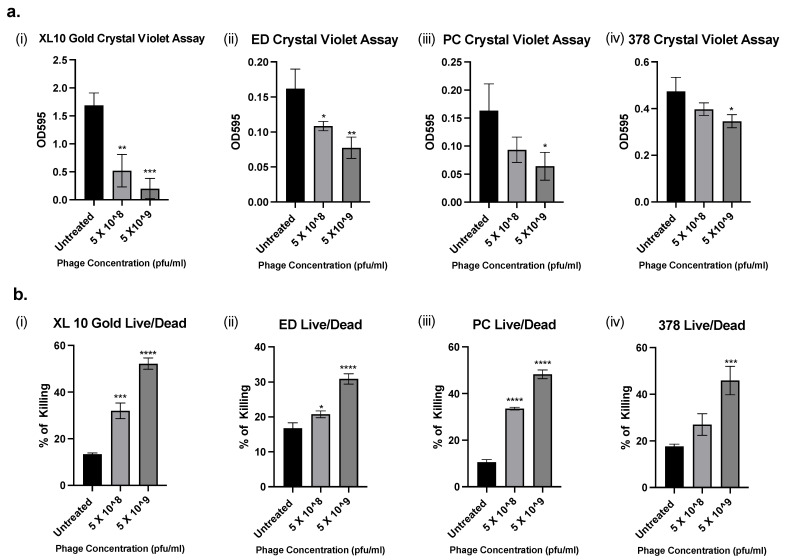
Assessment of biomass of different *E. coli* biofilms following bacteriophage treatment by (**a**) crystal violet and (**b**) Live/Dead assay. (**i**) XL10 Gold; (**ii**) ED (**iii**) PC and (**iv**) 378 *E. coli* biofilms were treated with phage APTC-EC-2A at a concentration of 5 × 10^8^ pfu/mL and 5 × 10^9^ pfu/mL. Biofilm biomass was measured (OD 595) and the percentage killing of *E. coli* biofilm with significance was determined compared to untreated control. Significance was determined by one-way ANOVA with Dunnett’s multiple comparisons test in comparison with untreated biofilms; *p* < 0.05 (*); *p* < 0.01 (**); *p* < 0.001 (***) and (****) *p* < 0.0001.

**Figure 6 microorganisms-10-00102-f006:**
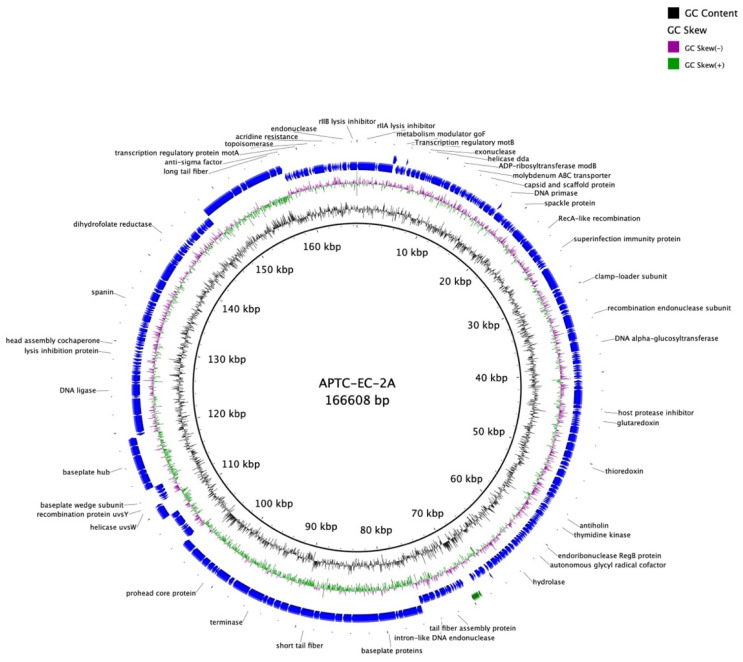
Map of the genomic organization of bacteriophage APTC-EC-2A. The Open Reading Frames with predicted annotations are indicated with blue arrows.

**Figure 7 microorganisms-10-00102-f007:**
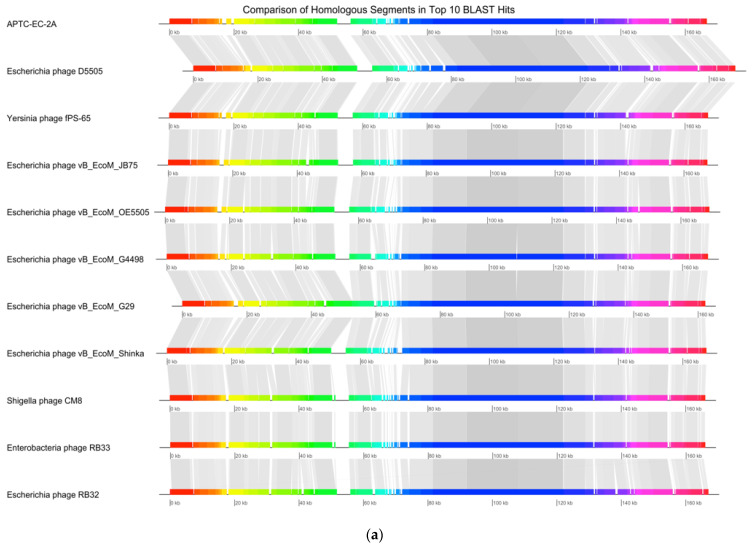
Phage Genome Sequence Analysis. (**a**) Mauve alignment of the annotated complete genomes of APTC-EC-2A with ten other related phages. The coloured segments represent the homologous nucleotide regions; whereas the gap regions indicate fragments that were not aligned or contained sequence elements specific to a particular genome. (**b**) A maximum-likelihood phylogenetic tree analysis of APTC-EC-2A with the top 25 hits phages based on the amino acid sequences of the phage terminase large subunits. APTC-EC-2A is highlighted in red.

**Table 1 microorganisms-10-00102-t001:** MIC for *E. coli* bacterial strains. MICs of seven bacterial strains were tested against 18 different types of antibiotics. Breakpoints used for testing *E. coli* strains were adopted from EUCAST.

Antimicrobial Class	Antimicrobial Agent	ECOFF ^a^ (µg/mL)	MIC (µg/mL)					
			378	452	ED	PC	RDI	OPS
Aminoglycosides	Gentamicin	2	>128	16	16	64	32	32
	Streptomycin	16	>128	>128	64	64	>128	>128
ß-lactam/ß-lactam inhibitor	Amoxicillin-clavulanate	-	128	64	16	16	128	32
Carbapenem	Ertapenem	-	1	<0.25	<0.25	<0.25	<0.25	<0.25
	Imipenem	0.001–4	2	2	4	2	4	4
Cephems	Cefoxitin	8	16	32	32	32	32	32
Fluoroquinolones	Ciprofloxacin	0.06	64	1	0.5	0.5	0.5	16
Folate pathway inhibitors	Trimethoprim	1	>128	>128	2	4	>128	>128
Penicillins	Ampicillin	8	>128	>128	8	8	>128	>128
Phenicol	Chloramphenicol	16	128	32	16	16	32	16
Tetracyclines	Doxycycline	8	8	16	8	8	8	64

^a^ EUCAST epidemiological cut-off values (µg/mL).

**Table 2 microorganisms-10-00102-t002:** Summary of *E. coli* phages isolation from Wastewater of TQEH. *E. coli* phage from wastewater from 5 different TQEH departments were screened. *E. coli* XL Gold was used as host. × = no *E. coli* phages; √ = *E. coli* phages present in the wastewater. ED = Emergency Department; PC = Palliative Care Ward; OPS = Orthopaedic and Plastic Surgery Ward; ICU = Intensive Care Unit; RDI = Respiratory, Dermatology and Infectious Ward; APTC = Adelaide Phage Therapy Centre.

Department of TQEH	Isolation Method	Presence of *E. coli* Phages	Phage Isolated
ED	Double-layer agar assay and Enrichment assay	×	
PC	Double-layer agar assay	√	APTC-EC-5, APTC-EC-6
OPS	Double-layer agar assay	√	APTC-EC-1, APTC-EC-3, APTC-EC-4
ICU	Double-layer agar assay	√	APTC-EC-2A, APTC-EC-2B
RDI	Double-layer agar assay	√	APTC-EC-7, APTC-EC-8A, APTC-EC-8B

**Table 3 microorganisms-10-00102-t003:** Morphological Characteristics of Isolated Phages.

Phage	Plaque Morphology	Titre (pfu/mL)
APTC-EC-1	Clear, ~3 mm	6.77 × 10^10^
APTC-EC-2A	Clear, ~2 mm	2.4 × 10^12^
APTC-EC-2B	Clear, ~2.5 mm	4.11 × 10^10^
APTC-EC-3	Clear, ~2 mm	4.7 × 10^9^
APTC-EC-4	Clear, ~2 mm	6.77 × 10^10^
APTC-EC-5	Clear, ~2 mm	2.3 × 10^9^
APTC-EC-6	Clear, ~2 mm	8.9 × 10^10^
APTC-EC-7	Clear, ~2.5 mm	3.8 × 10^8^
APTC-EC-8A	Clear, ~3 mm	4.2 × 10^12^
APTC-EC-8B	Clear, ~3 mm	1.07 × 10^8^

**Table 4 microorganisms-10-00102-t004:** Phage host range tests. Each phage (APTC-EC-1- APTC-EC-8B) host range was determined against 7 *E. coli* strains (XL10, ED 4, PC 22, RDI 14, OPS 13, 378, 452). + = clear plaque; +/− = turbid plaque; − = no lysis.

	APTC-EC-1	APTC-EC-2A	APTC-EC-2B	APTC-EC-3	APTC-EC-4	APTC-EC-5	APTC-EC-6	APTC-EC-7	APTC-EC-8A	APTC-EC-8B
XL10	+	+	+	+	+	+	+	+	+	+
ED 4	−	+	−	−	+/−	−	−	−	+/−	−
PC 22	−	+	−	−	−	−	−	−	+/−	+/−
RDI 14	−	+/−	−	−	−	−	−	−	−	+/−
OPS 13	−	−	−	−	−	−	−	−	−	−
378	−	+	−	−	−	−	−	−	+/−	+/−
452	−	+/−	−	−	−	−	−	−	−	−
No. of (semi)sensitive isolates	1	6	1	1	2	1	1	1	4	4

**Table 5 microorganisms-10-00102-t005:** BLAST Analysis. The top ten BLAST hits of APTC-EC-2A in the database.

Accession	Name	Percentage Identity
NC_054918	Escherichia phage vB_EcoM_G4498	98.97%
MK327940	Escherichia phage vB_EcoM_G29	98.41%
MK327929	Escherichia phage D5505	97.97%
NC_055724	Yersinia phage fPS-65	97.64%
NC_054926	Escherichia phage vB_EcoM_OE5505	97.40%
MZ502379	Escherichia phage vB_EcoM_Shinka	97.39%
MH355584	Escherichia phage vB_EcoM_JB75	97.38%
NC_054939	Shigella phage CM8	96.88%
KM607001	Enterobacteria phage RB33	96.29%
DQ904452	Escherichia phage RB32	96.29%

## Data Availability

Not applicable.

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
