# Peer review of "APTC-EC-2A: A Lytic Phage Targeting Multidrug Resistant E. coli Planktonic Cells and Biofilms"

_microorganisms, 2022, doi:10.3390/microorganisms10010102_

Round 1
Reviewer 1 Report
Review of ‘APTEC-ED-2A: a lytic phage with broad host range against multi-drug resistant E. coli
This research article presents interesting data on newly isolated E. coli phages and the authors conduct numerous experiments to showcase the phages lytic activity. The data is sound and overall the manuscript is written very well.
I have two comments, which apply to the whole manuscript:
- Add space before references and checking overall spacing in the manuscript as there are multiple gaps in the text.
- The authors describe the phage as ‘novel’ even though at the nucleotide level it shares 96-99% similarity to previously sequenced phages. Therefore, the phage is not novel and I suggest the authors change the manuscript accordingly.
Minor comments:
- Lines 36 to 37 – remove space before the commas
- Line 62 – add space before references
- Line 65 – add space before reference
- Line 116 – why is ul in blue? Also change ten to 10.
- Line 120 – remove extra spaces
- Line 132 – add reference to the double layer method
- Line 149 – add space before reference
- Line 163 – change ten to number 10
- Line 218 – pick between ‘phage’ and bacteriophage’ and then be consistent in the manuscript on which one you use
- Line 236 – remove extra spaces
- Figure 5a – the x axis titles are incorrect as you have two labelled as untreated. The x axis should be labelled as ‘untreated’, ‘5 x 10^8’ and ‘5 x 10^9’
- Line 381 – the line stating the phage is ‘novel’ is void as it has 96-99% similarity to other sequenced phages. I suggest the authors remove this sentence
- Line 390 – what do you mean by ‘spackle proteins’? Expand on this
Author Response
Comment 1: Add space before references and checking overall spacing in the manuscript as there are multiple gaps in the text.
Response 1: Thanks for your time and suggestions, we have revised the text according to your suggestions.
Comment 2: The authors describe the phage as ‘novel’ even though at the nucleotide level it shares 96-99% similarity to previously sequenced phages. Therefore, the phage is not novel and I suggest the authors change the manuscript accordingly.
Response 2: The word “novel” has been removed from the sentences and we have changed the manuscript accordingly.
Comment 3: Lines 36 to 37 – remove space before the commas
Response 3: Space removed, thanks!
Comment 4: Line 62 – add space before references
Response 4: Space added according to the comment.
Comment 5: Line 65 – add space before reference
Response 5: Space added according to the comment.
Comment 6: Line 116 – why is ul in blue? Also change ten to 10.
Response 6: Has been changed to black as well as the number.
Comment 7: Line 120 – remove extra spaces
Response 7: Space removed, thanks!
Comment 8: Line 132 – add reference to the double layer method
Response 8: Reference for double layer method has been referenced earlier in line 113
Comment 9: Line 149 – add space before reference
Response 9: Space added according to the comment.
Comment 10: Line 163 – change ten to number 10
Response 10: Changes made according to the comment.
Comment 11: Line 218 – pick between ‘phage’ and bacteriophage’ and then be consistent in the manuscript on which one you use
Response 11: We have changed bacteriophage to phage to be consistent.
Comment 12: Line 236 – remove extra spaces
Response 12: Space removed, thanks!
Comment 13:Figure 5a – the x axis titles are incorrect as you have two labelled as untreated. The x axis should be labelled as ‘untreated’, ‘5 x 10^8’ and ‘5 x 10^9’
Response 13: X-axis has been labelled correctly.
Comment 14: Line 381 – the line stating the phage is ‘novel’ is void as it has 96-99% similarity to other sequenced phages. I suggest the authors remove this sentence
Response 14: The word “novel” has been removed from the sentences.
Comment 15: Line 390 – what do you mean by ‘spackle proteins’? Expand on this
Response 15: Discussion on spackle proteins has been added to the discussion part, line 526
Thank you for the time and effort spent by the reviewers on our manuscript, and we hope that our explanations and modifications will address the concerns raised.
Reviewer 2 Report
Hon et al. presents a manuscript entitled "APTC-EC-2A: A lytic phage with broad host range against multidrug-resistant E. coli" manuscript is generally correct but I have some comments.
with respect to methodology, why authors used only 7 e.coli strains? You should contact with microbiology dep. and get more strains to do a proper host range of isolated phages. if not, You should present phylogenetics of strains used to show their non-relativenes (You can analyze MALDI data with R programing to show that).
in paragraph "2.6. purification of phage" the symbol uL has a hyperlink to the wiktionary.
data of resistance should be in main text, not in supplementary due to the title of Yor work. additionally, should be after bacterial isolation paragraph.
Authors satisfactory discuss obtained data but in this work is lack of perspective - You should indicate the use of phages against E.coli like: treatment and microbiota control (https://www.frontiersin.org/articles/10.3389/fmicb.2019.01984/full) food antimicrobial protection (https://doi.org/10.1111/jfs.12747 10.4161/bact.22825 ) or environmental control (https://doi.org/10.1016/j.apsoil.2017.06.020)
Author Response
Comment 1: with respect to methodology, why authors used only 7 e.coli strains? You should contact with microbiology dep. and get more strains to do a proper host range of isolated phages. if not, You should present phylogenetics of strains used to show their non-relativenes (You can analyze MALDI data with R programing to show that).
Response 1: Here we have tested the isolated phage’s sensitivity toward 7 E. coli strains. Those 7 E. coli strains were isolated either from the wastewater of QEH (various departments and multiple time points) or a Nursing home (isolates donated by Dr Rietie Venter), and all were multi-drug resistant according to our MIC result. The main aim of this study was to isolate lytic phages that specifically kill those multi-drug resistant strains. Given that 6/7 MDR E. coli strains tested could be lysed by APTC-EC-2A, and that those strains were isolated from various departments and at multiple time points, we postulate that their relatedness will be diverse. However, we do agree that in order to make the claim that the phage has a broad host range, further testing is required. For this, we will do phage susceptibility testing of further E. coli strains and we will also sequence those strains to determine their relatedness. Given the limited time that we are given to file this revised version of this paper, results will be presented in a follow-up paper where further phages and their activity will be disclosed. To help address the concern, however, we have removed the claim of APTC-EC-2A having a broad host range in the title and in the abstract.
Comment 2: data of resistance should be in main text, not in supplementary due to the title of your work. additionally, should be after bacterial isolation paragraph.
Response 2: Data of resistance has been moved to the main text after the bacterial isolation paragraph and is now labelled as Table 1.
Comment 3: Authors satisfactory discuss obtained data but in this work is lack of perspective - You should indicate the use of phages against E.coli like: treatment and microbiota control (https://www.frontiersin.org/articles/10.3389/fmicb.2019.01984/full) food antimicrobial protection (https://doi.org/10.1111/jfs.12747 10.4161/bact.22825 ) or environmental control (https://doi.org/10.1016/j.apsoil.2017.06.020)
Response 3: Thanks for your suggestion. Indeed, the phage has been used against E. coli in different areas, such as food industries or food safety. We have addressed this in the discussion section and added the references as requested.
Reference 36 : Połaska, M. and B. Sokołowska, Bacteriophages-a new hope or a huge problem in the food industry. AIMS Microbiol, 2019. 5(4): p. 324-346.
Reference 37: Sillankorva, S.M., H. Oliveira, and J. Azeredo, Bacteriophages and their role in food safety. Int J Microbiol, 2012. 2012: p. 863945.
Thank you for the time and effort spent by the reviewers on our manuscript, and we hope that our explanations and modifications will address the concerns raised
Round 2
Reviewer 2 Report
NA